# Evidence for transmission of COVID-19 prior to symptom onset

**Lauren C Tindale[1†], Jessica E Stockdale[2†], Michelle Coombe[1], Emma S Garlock[2], Wing Yin Venus Lau[2], Manu Saraswat[1], Louxin Zhang[3], Dongxuan Chen[4,5], Jacco Wallinga[4,5], Caroline Colijn[2]***

[1]University of British Columbia, Vancouver, Canada; [2]Simon Fraser University, Burnaby, Canada; [3]National University of Singapore, Singapore, Singapore; [4]Centre for Infectious Disease Control, National Institute for Public Health and the Environment, Bilthoven, Netherlands; [5]Leiden University Medical Center, Leiden, Netherlands

**Abstract** We collated contact tracing data from COVID-19 clusters in Singapore and Tianjin, China and estimated the extent of pre-symptomatic transmission by estimating incubation periods and serial intervals. The mean incubation periods accounting for intermediate cases were 4.91 days (95%CI 4.35, 5.69) and 7.54 (95%CI 6.76, 8.56) days for Singapore and Tianjin, respectively. The mean serial interval was 4.17 (95%CI 2.44, 5.89) and 4.31 (95%CI 2.91, 5.72) days (Singapore, Tianjin). The serial intervals are shorter than incubation periods, suggesting that pre-symptomatic transmission may occur in a large proportion of transmission events (0.4–0.5 in Singapore and 0.6–0.8 in Tianjin, in our analysis with intermediate cases, and more without intermediates). Given the evidence for pre-symptomatic transmission, it is vital that even individuals who appear healthy abide by public health measures to control COVID-19.

**\*For correspondence:**
ccolijn@sfu.ca

[†]These authors contributed equally to this work

## Introduction

The novel coronavirus disease, COVID-19, was first identified in Wuhan, Hubei Province, China in December 2019 (*Li et al., 2020b*; *Huang et al., 2020*). The virus causing the disease was soon named severe acute respiratory syndrome coronavirus 2 (SARS-CoV-2) (*Hui et al., 2020*) and quickly spread to other regions of China and then across the globe, causing a pandemic with over 5 million cases and 300,000 deaths at the time of writing (*Johns Hopkins University, 2020*). In Tianjin, a metropolis located at the north of China, the first case was confirmed on January 21, 2020 (*Tianjin Health Commission, 2020*). Two days later, the first case was confirmed in Singapore (*Ministry of Health Singapore, 2020*), a city country in Southeast Asia. As of February 28, 2020, 93 and 135 cases had been confirmed in Singapore and Tianjin (*Ministry of Health Singapore, 2020*; *Tianjin Health Commission, 2020*). The first Singapore COVID-19 case was confirmed as an individual who had travelled to Singapore from Wuhan. Many of the initial cases were imported from Wuhan, with later cases being caused by local transmission. Singaporean officials worked to identify potential contacts of confirmed cases; close contacts were monitored and quarantined for 14 days from their last exposure to the patient, and other low-risk contacts were put under active surveillance and contacted daily to monitor their health status. These early outbreaks continue to provide the opportunity to estimate key parameters to understand COVID-19 transmission dynamics.

We screened publicly available data to identify datasets for two COVID-19 clusters that could be used to estimate transmission dynamics. In both Singapore and Tianjin, the COVID-19 outbreak occurred within a relatively closed system where immediate public health responses were implemented, contacts were identified and quarantined, and key infection dates were tracked and updated daily. With its experiences in control of the SARS outbreak, the Singaporean government

**eLife digest** The first cases of COVID-19 were identified in Wuhan, a city in Central China, in December 2019. The virus quickly spread within the country and then across the globe. By the third week in January, the first cases were confirmed in Tianjin, a city in Northern China, and in Singapore, a city country in Southeast Asia. By late February, Tianjin had 135 cases and Singapore had 93 cases. In both cities, public health officials immediately began identifying and quarantining the contacts of infected people.

The information collected in Tianjin and Singapore about COVID-19 is very useful for scientists. It makes it possible to determine the disease's incubation period, which is how long it takes to develop symptoms after virus exposure. It can also show how many days pass between an infected person developing symptoms and a person they infect developing symptoms. This period is called the serial interval. Scientists use this information to determine whether individuals infect others before showing symptoms themselves and how often this occurs.

Using data from Tianjin and Singapore, Tindale, Stockdale et al. now estimate the incubation period for COVID-19 is between five and eight days and the serial interval is about four days. About 40% to 80% of the novel coronavirus transmission occurs two to four days before an infected person has symptoms. This transmission from apparently healthy individuals means that staying home when symptomatic is not enough to control the spread of COVID-19. Instead, broad-scale social distancing measures are necessary.

Understanding how COVID-19 spreads can help public health officials determine how to best contain the virus and stop the outbreak. The new data suggest that public health measures aimed at preventing asymptomatic transmission are essential. This means that even people who appear healthy need to comply with preventive measures like mask use and social distancing.

had been adopting a case-by-case control policy from January 2, 2020. Only close contacts of a confirmed case were monitored and quarantined for 14 days. In Tianjin, a number of COVID-19 cases were traced to a department store, where numerous customers and sales associates were likely infected. Additional customers who had potential contact were asked to come forward through state news and social media, as well as asked if they had visited the department store at various checkpoints in the city. All individuals identified as having visited the store in late January were quarantined and sections of the Baodi District where the store is located were sealed and put under security patrol.

We estimate the serial interval and incubation period of COVID-19 from clusters of cases in Singapore and Tianjin. The serial interval is defined as the length of time between symptom onset in a primary case (infector) and symptom onset in a secondary case (infectee), whereas the incubation period is defined as the length of time between an infectee's exposure to a virus and their symptom onset. Both are important parameters that are widely used in modeling in infectious disease, as they impact model dynamics and hence fits of models to data. While the pandemic has progressed far beyond these early outbreaks, it remains the case that mathematical modelling, using parameters derived from estimates like these, is widely used in forecasting and policy.

The serial interval and incubation period distributions, in particular, can be used to identify the extent of pre-symptomatic transmission (i.e. viral transmission from an individual that occurs prior to symptom onset). There is evidence that pre-symptomatic transmission accounts for a considerable portion of COVID-19 spread (*Arons et al., 2020*; *Baggett et al., 2020*; *Li et al., 2020a*) and it is important to determine the degree to which this is occurring (*Peak et al., 2020*). Early COVID-19 estimates borrowed parameters from SARS (*Wu et al., 2020*; *Jiang et al., 2020*; *Abbott et al., 2020*), but more recent estimates have been made using information from early clusters of COVID-19 cases, primarily in Wuhan. Depending on the population used, estimates for incubation periods have ranged from 3.6 to 6.4 days and serial intervals have ranged from 4.0 to 7.5 days (*Li et al., 2020b*; *Ki and Task Force for 2019-nCoV, 2020*; *Backer et al., 2019*; *Linton et al., 2020*; *Nishiura et al., 2020*); however, it is crucial that the estimates of incubation period and serial interval are based on the same outbreak, and are compared to those obtained from outbreaks in other populations. Distinct outbreak clusters are ideal for understanding how COVID-19 can spread through a

population with no prior exposure to the virus. Here, we estimate the portion of transmission that is pre-symptomatic based on estimates of the incubation period and serial interval. We estimate both quantities under two frameworks: first, we use samples as directly as is feasible from the data, for example assuming that the health authorities' epidemiological inferences regarding who exposed whom and who was exposed at which times are correct. Second, we use estimation methods that allow for unknown intermediate cases, such that the presumed exposure and infection events may not be complete. We also separate the analysis of incubation period according to earlier and later phases of the outbreaks, since measures were introduced during the time frame of the data.

## Results

### Descriptive analyses

*Figures 1* and *2* show the daily counts, putative origin of the exposure and individual time courses for the Singapore and Tianjin data. In the Singapore dataset, new hospitalization and discharge

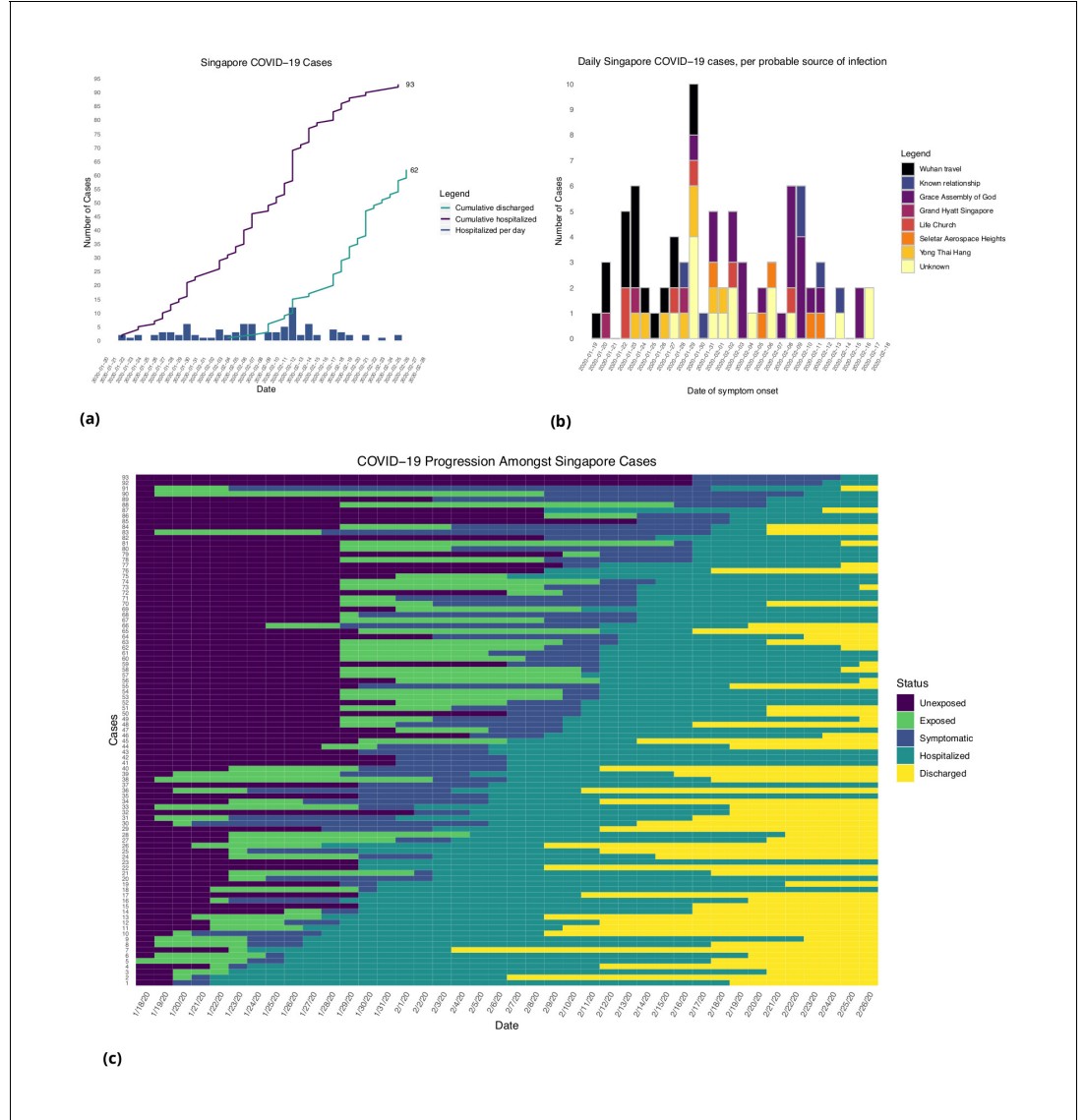

**Figure 1.** Singapore COVID-19 cases. (a) Daily hospitalized cases and cumulative hospitalized and discharged cases. (b) Daily incidence with probable source of infection. (C) Disease timeline, including dates at which each case is unexposed, exposed, symptomatic, hospitalized, and discharged. Not all cases go through each status as a result of missing dates for some cases.

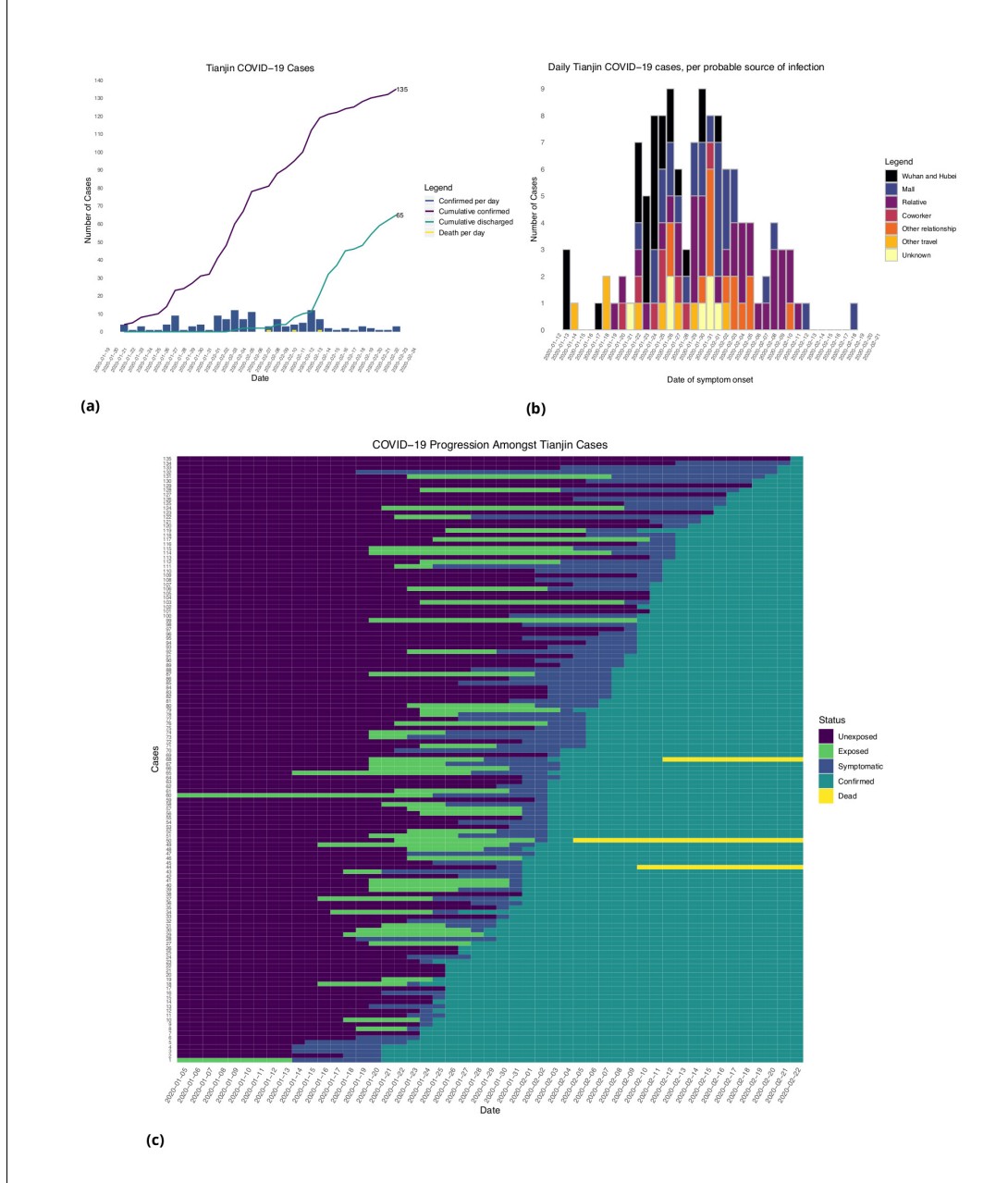

**Figure 2.** Tianjin COVID-19 cases. (**a**) Daily and cumulative confirmed cases, cumulative discharges and daily death cases. (**b**) Daily incidence with probable source of infection. (**c**) Disease progression timeline; not all cases go through each status as a result of missing dates for some cases.

cases were documented daily from January 23 to February 26, 2020. 66.7% (62/93) of the confirmed cases recovered and were discharged from the hospital by the end of the study period (*Figure 1(a)*). The disease progression timeline of the 93 documented cases in *Figure 1(c)* indicates that symptom onset occurred 1.71 ± 3.01 (mean ± SD) days after the end of possible viral exposure window and cases were confirmed 7.43 ± 5.28 days after symptom onset. The mean length of hospital stay was 13.3 ± 6.01 days before individuals recovered and were discharged.

In the Tianjin dataset, new confirmed cases were documented daily from January 21 to February 22, 2020. 48.1% (65/135) recovered and 2.2% (3/135) had died by the end of the study period (*Figure 2(a)*). The timeline of the 135 cases is shown in *Figure 2(c)*. Symptom onset occurred 4.98 ± 4.83 (mean ± SD) days after the end of the possible viral exposure window. Cases were confirmed 5.23 ± 4.15 days after symptom onset. The duration of hospital stay of the Tianjin cases is unknown

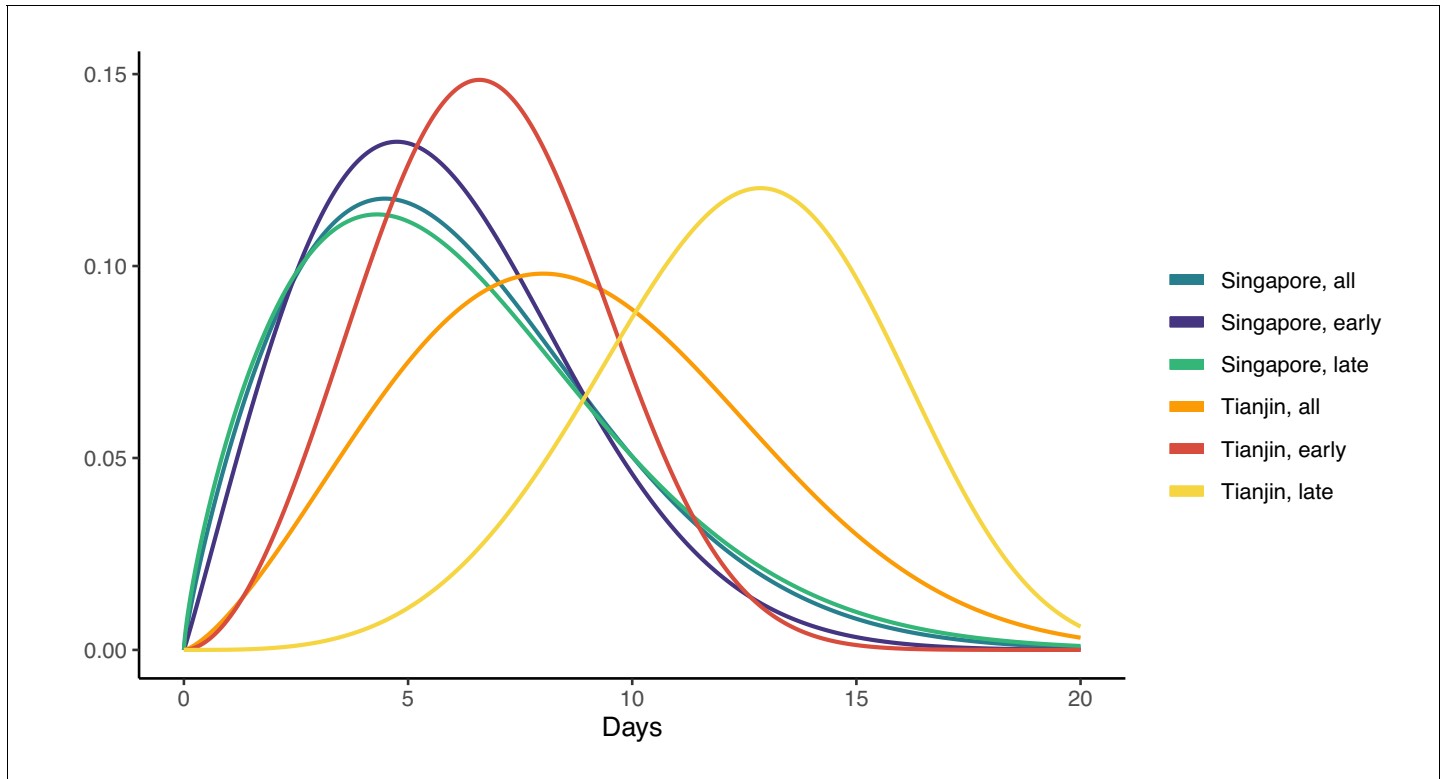

**Figure 3.** Fitted gamma COVID-19 incubation period distributions (without intermediates). Cases are defined as 'early' if they have symptom onset on or prior to January 31, and are classified 'late' otherwise.

**Table 1.** Mean incubation period, serial interval and pre-symptomatic transmission.
Incubation periods are based on the gamma estimates because these are the most convenient for taking the covariation of serial intervals and incubation periods into account (done throughout the table). 95% CIs are provided in brackets.

| Without intermediates | Incubation (days) | Serial interval (days) | Mean difference (days) | Portion pre-symptomatic(-) |
|---|---|---|---|---|
| Singapore (all) | 5.99 (4.97, 7.14) | 4.0 (2.73, 5.57) | 1.99 | 0.74 |
| Singapore (early) | 5.91 (4.50,7.64) | | 1.91 | 0.742 |
| Singapore (late) | 6.06 (4.70, 7.67 ) | | 2.06 | 0.744 |
| Tianjin (all) | 8.68 (7.72, 9.7) | 5.0 (3.82, 6.12) | 3.68 | 0.81 |
| Tianjin (early) | 6.88 (5.97,7.87) | | 1.88 | 0.72 |
| Tianjin (late) | 12.4 (11.1,13.7) | | 7.4 | 0.96 |
| **Account for intermediates** | | | | |
| Singapore $r = 0.05$ | 4.91 | 4.17 (2.44, 5.89) | 0.77 | 0.53 |
| Singapore $r = 0.1$ | 4.43 | | 0.26 | 0.46 |
| Singapore $r = 0.15$ | 4.12 | | −0.05 | 0.41 |
| Singapore $r = 0.2$ | 3.89 | | −0.28 | 0.38 |
| Tianjin $r = 0.05$ | 7.54 | 4.31 (2.91, 5.72) | 3.23 | 0.79 |
| Tianjin $r = 0.1$ | 6.89 | | 2.58 | 0.74 |
| Tianjin $r = 0.15$ | 6.30 | | 1.99 | 0.67 |
| Tianjin $r = 0.2$ | 5.91 | | 1.6 | 0.64 |

as the discharge date of each case was not available. In both datasets, daily counts decline over time, which is likely a combination of delays to symptom onset and between symptom onset and reporting, combined with the effects of strong social distancing and contact tracing.

## Incubation period

In the Singapore dataset, we find that the median incubation period in our direct analysis (without accounting for intermediate cases) is 5.32 days with the gamma distribution; shape 3.05 (95%CI 2.0, 3.84); and scale 1.95 (1.23, 2.34). The mean incubation period is 5.99 (95%CI 4.97, 7.14) days. In Tianjin, we find a median 8.06 days; shape 4.74 (3.35, 5.72); scale 1.83 (1.29, 2.04). The mean is 8.68 (7.72, 9.7) days. These results are summarised in *Table 1*, and we also fitted Weibull and log normal distributions; see *Appendix 1—table 1*. These are consistent with, or slightly longer than, previous estimates, see *Appendix 1—table 5* for comparison.

In Singapore, these estimates are based on a combination of cases for whom last possible exposure is given by travel, and later cases (for whom the presumed infector was used). In Tianjin, social distancing measures were implemented during the outbreak. We find that the estimated incubation period is different, particularly in Tianjin, for cases with symptom onset on or prior to January 31st: see *Figure 3* and *Figure 4*. The estimated median incubation period for pre-Feb one cases in Tianjin is 6.48 days; the $q = (0.025, 0.975)$ quantiles are (2.5, 13.3) days. In contrast, post-Jan 31 the median is 12.13 days with $q = (0.025, 0.975)$ quantiles (7.3, 18.7) days. The means are 6.88 (5.97, 7.87) days for early cases and 12.4 (11.1, 13.7) days for later cases. Social distancing seems unlikely to change the natural course of infection, but these results might be explained if exposure occurred during group quarantine or otherwise later than the last time individuals thought they could have been exposed. Pre-symptomatic transmission would enable this, if an individual was thought to have been exposed before group quarantine, but in actuality was exposed during quarantine by a pre-

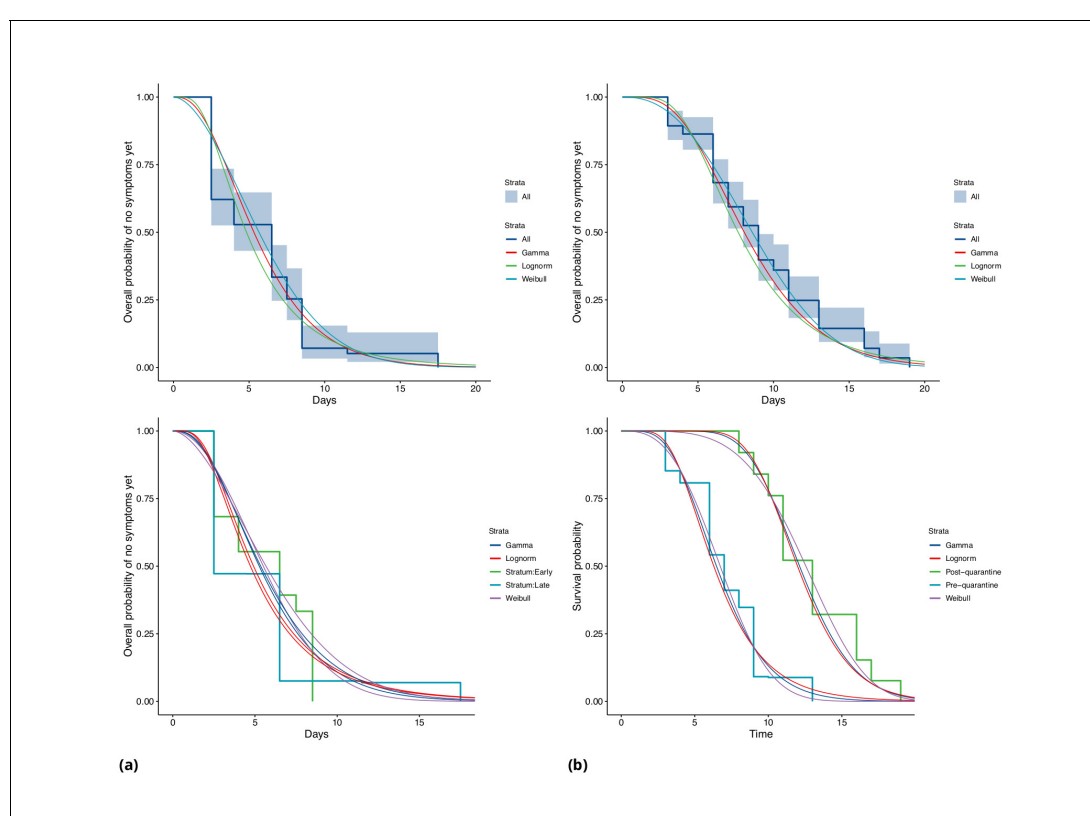

**Figure 4.** COVID-19 incubation period Kaplan-Meier curves for (a) Singapore and (b) Tianjin. Top panels show unstratified data (all cases with symptom onset given). Bottom panels show 'early' and 'late' cases, where early cases are defined as those with symptom onset on or prior to January 31, and late otherwise.

symptomatic individual. The time interval in the data would then not be a sample of the incubation period, instead it would be a sample of one or more generation times plus an incubation period.

In Singapore, we find the same effect, although much less pronounced. The estimated median incubation time is 5.26, with (0.025, 0.975) quantiles of (1.30, 13.8) days for early cases (also defined as cases with symptom onset on or prior to January 31st) and 5.35 (quantiles (1.22, 14.6)) days for late-arising cases. The means are 5.91 (4.50, 7.64) days for early cases and 6.06 (4.70, 7.67) days for later cases. Fits of gamma and log-normal distributions are similar; see *Appendix 1—table 2*. Changes in perception of exposure times after control measures were introduced (i.e. people may assume that they must have been exposed prior to control measures), together with pre-symptomatic transmission, could result in missing intermediate transmission events and hence lengthened incubation period estimates. This in part motivates our analysis with intermediate cases.

Our estimates of the incubation period with intermediates are similar, under the assumption that intermediates are relatively rare. Results are shown in *Figure 5* and *Table 1*. We find that the median of the bootstrapped mean incubation periods for Singapore with a low (0.05 per day) rate of unknown intermediates is 4.91 days (4.35, 5.69 95% bootstrap CI), compared to a generation time of 3.71 (2.36, 4.91) days. The Tianjin bootstrapped mean incubation period is 7.54 (6.76, 8.56 95% CI) days and the generation time is only 2.82 (1.83, 3.52) days. The estimates are lower when the assumed probability of unknown intermediates is higher. Indeed, if intermediates were present

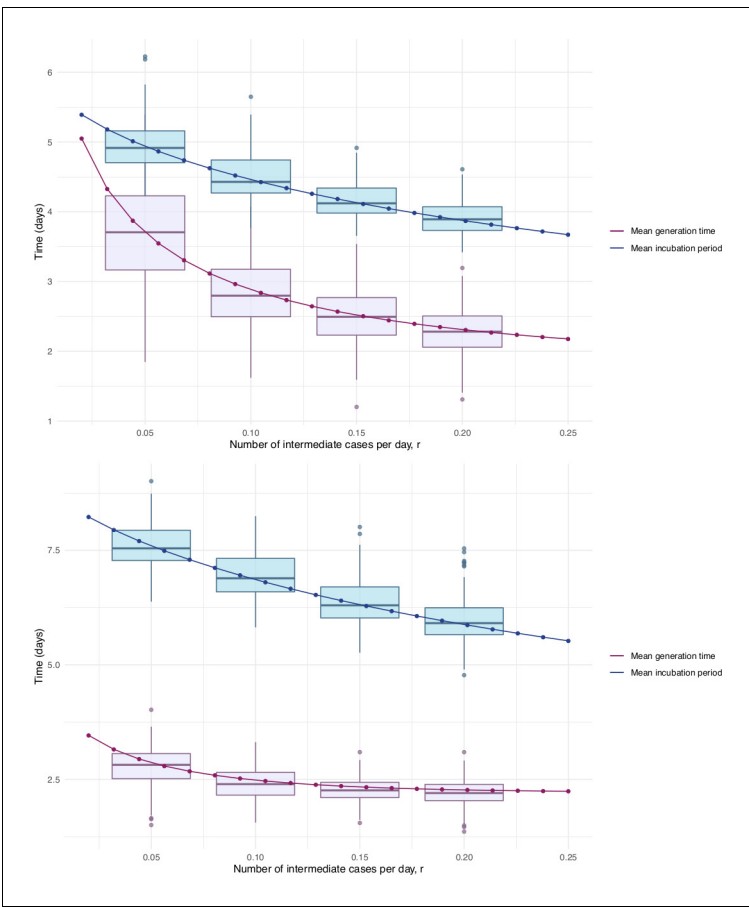

**Figure 5.** Mean incubation period and generation time estimates from the incubation period intermediates analysis, under the assumption that the scale parameter for both distributions is equal, shown with dependence on the mean number of unknown intermediate cases per day of the empirical time elapsed between exposure and symptom onset. The incubation period is longer than the generation time, so this analysis suggests that symptom onset occurs after infectiousness begins. Top: Singapore. Bottom: Tianjin. The means are the scale times the shape, which is fixed at 2.1 in Singapore and 2.2 in Tianjin. Varying this fixed value for the shape parameter was not found to significantly impact the results.

between assumed exposure and onset, naturally the generation time would be shorter than if they were not. The mean generation times are consistently shorter than the mean incubation periods, indicating that infection can occur prior to symptom onset. The difference is particularly pronounced in Tianjin, where long intervals were observed.

However, this approach makes a number of assumptions and is limited by the fact that if we do not know the true infectors then we are also unlikely to know the true exposure. The data we have is well suited to this method in the sense that there were particular events where exposure is thought to have occurred, and so we can account for intermediates in the manner we have done, but we do not have information for the alternative scenario in which the true exposures were prior to those given in the data. This could happen if, for example, individuals were exposed before attending an event or before known contact, and developed symptoms well after it. Exposure would thus be wrongly attributed to the event or contact. We have accommodated this with uncertainty in the exposure intervals, in particular not insisting that individuals who are likely to be the index case for a cluster (e.g. who developed symptoms on the same day as an event) *must* have been exposed then, but instead allowing the possibility that they were exposed earlier.

## Serial intervals

*Figure 6* represents the empirical serial intervals between all potential transmission case-pairs as noted in the data and represented in *Figure 7*, split into groups based on date of first symptom onset for each case-pair. The empirical mean serial intervals shorten in the 'late' group in both Singapore and Tianjin; however, the empirically derived 95% confidence intervals overlap (Singapore early 4.44 (-2.81, 11.7) vs. late 3.18 (-1.52, 7.88); Tianjin early 5.48 (-0.968, 11.9) vs. late 4.18 (-2.33, 10.7)). Negative lower bounds are due to the high standard deviation.

Shortening serial intervals are expected as increased quarantine measures are enacted during the course of an outbreak and can be an indication of improved control through successful contact tracing, as seen in SARS (*Lipsitch et al., 2003*). Our results suggest that serial intervals shortened as the outbreak progressed in both clusters, but they could also be due to right truncation. Accounting for

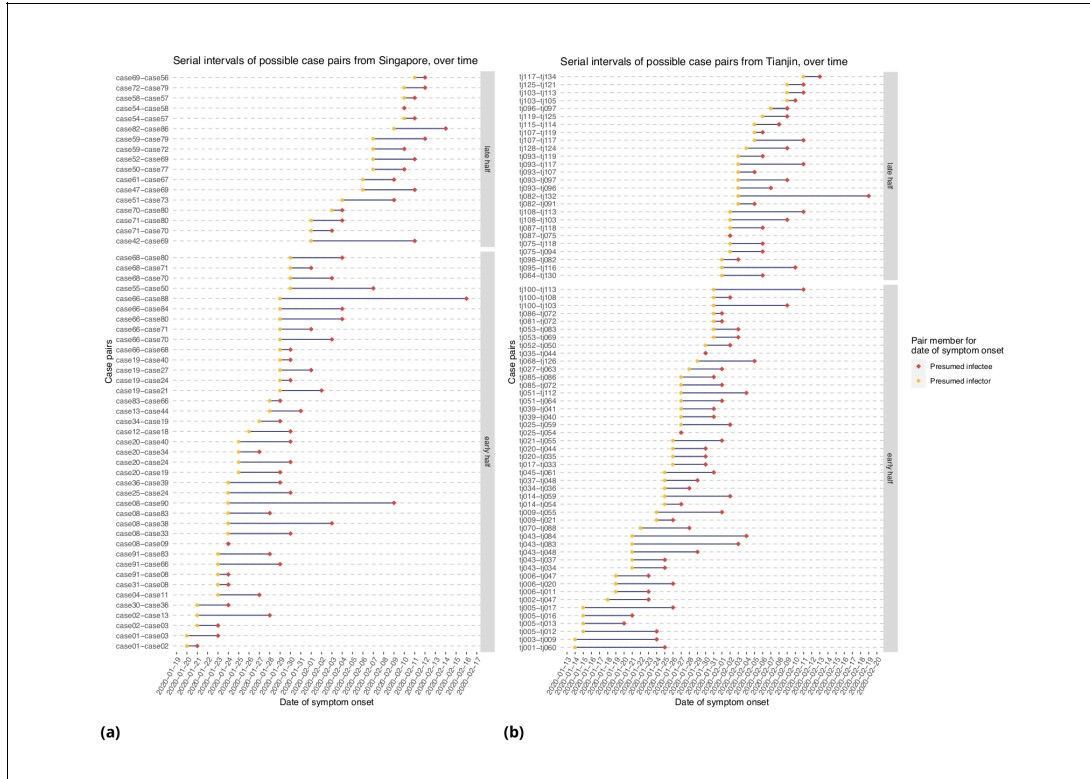

**Figure 6.** Serial intervals of possible case pairs in (a) Singapore and (b) Tianjin. Pairs represent a presumed infector and their presumed infectee plotted by date of symptom onset. Cases are defined as 'early' if they have symptom onset on or prior to January 31st.

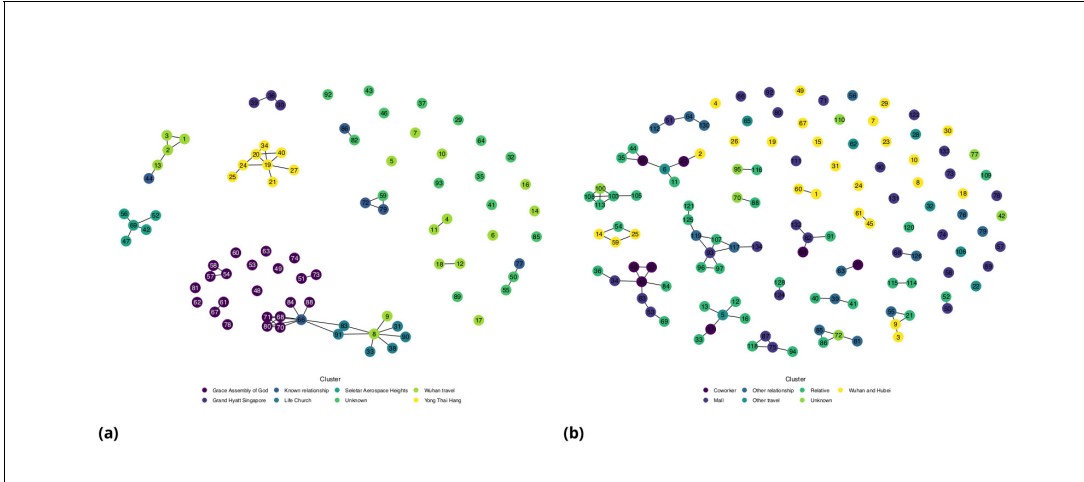

**Figure 7.** Network diagram for (**a**) Singapore (**b**) Tianjin.

this, we found that the mean serial intervals were 4 and 5 days (Singapore, Tianjin); a Cox regression found no significant difference between the early and late groups' serial intervals. This estimate is made directly from case pairs in the data without accounting for intermediate infectors and co-primary infection, as in the ICC analysis.

*Table 1* shows our ICC estimates of the mean and standard deviation for the serial intervals, with comparison to other analyses and assumptions in *Appendix 1—table 5*. The ICC method finds the mean serial interval to be 4.17 (2.44, 5.89 95% bootstrap CI) days (0.882 bootstrap standard deviation) for Singapore and 4.31 (2.91, 5.72) days (0.716 bootstrap sd) for Tianjin, using the first four cases in each cluster. This is consistent with the results with right truncation.

## Pre-symptomatic transmission

We estimated incubation periods and serial intervals with and without accounting for intermediate unknown cases. To estimate the portion of transmission that occurs before symptom onset, we compare the 'direct' (no intermediate) estimates of each, and the 'indirect' (accounting for intermediates) estimates of each. We estimate consistently shorter serial intervals than incubation period, suggesting that there is pre-symptomatic transmission.

We took the covariation of incubation periods and serial intervals (and of generation times and incubation periods) into account by sampling the intervals jointly before estimating the fraction of the relevant differences that are negative. Even accounting for correlation, the estimated fraction of pre-symptomatic transmission for Singapore is 0.74 (regardless of early/late split) and for Tianjin is 0.72, 0.96, 0.81 (early, late, all), based on the direct estimates of the incubation periods and serial intervals (see also *Figure 8*). When we use the incubation period estimates that account for intermediates, the portions pre-symptomatic transmission are 0.53 in Singapore and 0.79 in Tianjin, when the assumed 'rate of appearance' of intermediates $r$ is 0.05 (i.e. when we assume a relatively low rate of unknown intermediates). If this rate $r$ is increased, the portion of pre-symptomatic transmission decreases, but even for $r = 0.2$ we estimate the pre-symptomatic portion to be 0.38 in Singapore and 0.64 in Tianjin.

These results were obtained under an estimated correlation between the incubation period and serial interval of 0.289 in Tianjin. If instead the correlation were 0.1, the portion of pre-symptomatic transmission in Tianjin under $r = 0.05, 0.1, 0.15$ and 0.2, respectively, is estimated as 0.783, 0.725, 0.663 and 0.62. With correlation 0.8, the equivalent portions are 0.849, 0.781, 0.704 and 0.660. We therefore find that the degree of positive correlation does not greatly impact our estimates of pre-symptomatic transmission. We retain high estimates of the fraction pre-symptomatic in Tianjin, due to the long apparent incubation periods. It seems likely that these are an artifact of either pre-symptomatic transmission during quarantine/lockdown, or of other assumptions made about exposures in the creation of the original dataset. We conclude that overall for this data and under reasonable assumptions, we see evidence of at least 65% of transmission occurring before symptom onset.

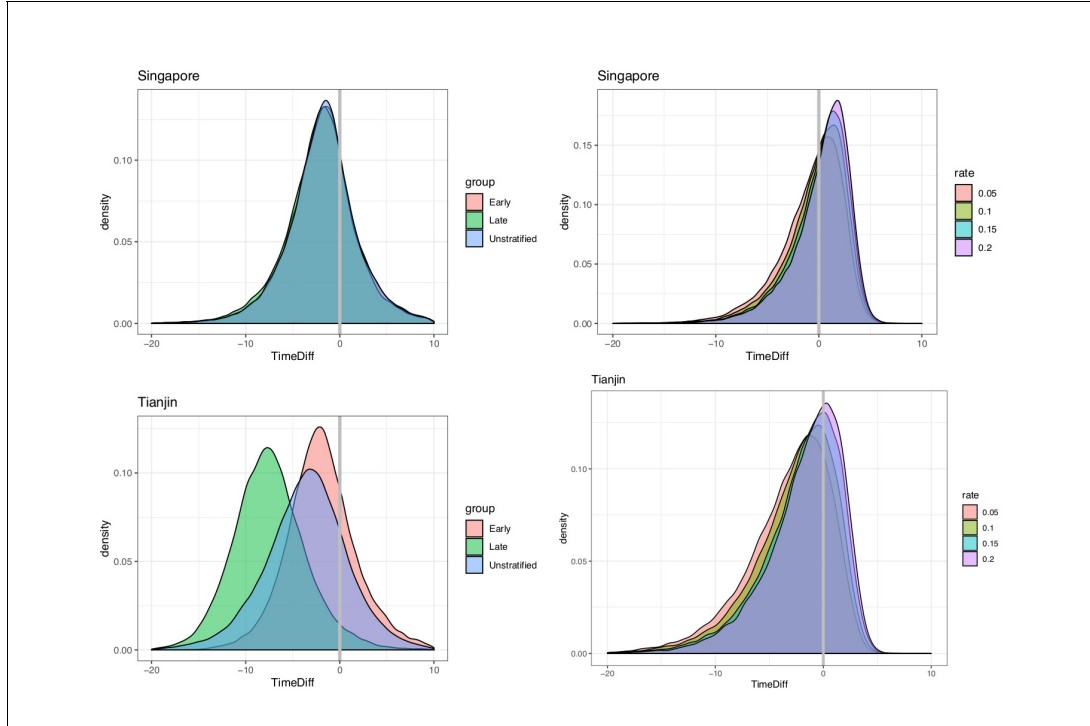

**Figure 8.** Pre-symptomatic infection as estimated by samples of (serial interval - incubation period), accounting for covariation. Top: Singapore. Bottom: Tianjin. Left: without intermediates. Right: accounting for intermediates. Grey vertical line: 0. Samples below zero indicate pre-symptomatic transmission. In all cases there is substantial pre-symptomatic transmission.

In our direct analysis, we estimate that infection occurred on average 1.99 and 3.68 days before symptom onset of the infector (Singapore, Tianjin). Because the incubation period is different for early- and late-occurring cases in our data, on average transmission for early-occurring cases is 1.91 and 2.06 days before symptom onset (Singapore, Tianjin) and 1.88, 7.4 days before (Singapore, Tianjin) for late-occurring cases. Taking a low rate ($r = 0.05$) of potential unknown intermediate cases into account, the mean difference reduces to 0.77 and 3.23 days (Singapore, Tianjin), though we still estimate a significant portion of pre-symptomatic transmission (0.53, 0.79), as above.

Overall, serial intervals are robustly shorter than incubation periods in our analyses (*Table 1*). These estimates are strengthened by the fact that we have estimated both incubation period and serial interval in the same populations and by the fact that we obtain the same result in two distinct datasets. In both sets of estimates, samples of the incubation period minus serial interval are negative with probability 0.38 or higher (Singapore) and 0.64 or higher (Tianjin), and these lower bounds require a high rate of unknown intermediates early in the outbreak. This indicates that a substantial portion of transmission may occur before symptom onset (see Appendix 1 and *Figure 8*), consistent with the clinical observations reported by *Rothe et al., 2020* and (*Bai et al., 2020*).

Shorter serial intervals yield lower reproduction number estimates. For example, if the epidemic grows at a rate of 0.15 (doubling time of 4.6 days [*Jung et al., 2020*], scenario 1), an estimated reproduction number using the mean of the bootstrapped estimates is $R = 1.76 \sim (1.30, 2.17)$ with a serial interval of 4.17 days (Singapore) and $R = 1.95 \sim (1.72, 2.47)$ with a serial interval of 4.3 days (Tianjin). In contrast, if a longer serial interval (7.5 days [*Jung et al., 2020*; *Li et al., 2020b*]) is used, the estimate is $R = 3.05$. This is based on the relationship between $R0$, serial interval, and growth rate, and is a simple estimate that does not take into account a complex and variable natural history of infection (*Wallinga and Lipsitch, 2007*). It serves primarily to illustrate how our estimated serial intervals impact $R$ in simple models for COVID-19 dynamics.

## Discussion

Here, we use transmission clusters in two locations where cases have reported links, exposure and symptom onset times to estimate both the incubation period and serial interval of COVID-19. We make these datasets available in a convenient spreadsheet form; they were available publicly but the Singapore dataset was presented in free text updates and the Tianjin cluster was described on multiple sites and in graphical form, in Chinese. We anticipate that the datasets themselves will remain useful for understanding COVID-19's early spread in these well-documented outbreaks.

The incubation period and serial interval are key parameters for transmission modeling and for informing public health interventions; modeling remains one of the primary policy aids in use in planning local and global COVID-19 responses. Serial intervals, together with R0, control the shape and distribution of the epidemic curve (*Anderson et al., 2004*). They influence the disease's incidence and prevalence, how quickly an epidemic grows, and how quickly intervention methods need to be implemented by public health officials to control the disease (*Anderson et al., 2004*; *Fraser et al., 2004*). In particular, the portion of transmission events that occur before symptom onset is a central quantity for infection control (*Fraser et al., 2004*), and will impact the efficacy of contact tracing and case finding efforts (*Peak et al., 2020*).

Singapore and Tianjin officials both reacted quickly when COVID-19 cases appeared and started implementing contact tracing and containment measures; however, there was a dramatic difference in the severity of the measures taken. The first case was identified in Singapore on Jan 23, 2020 and in Tianjin on Jan 21. By Feb 9, Singapore had identified 989 close contacts and implemented a travel advisory to defer all travel to Hubei Province and all non-essential travel to Mainland China, asked travellers to monitor their health closely for 2 weeks upon return to Singapore, and asked the public to adopt precautions including avoiding close contact with people who are unwell, practicing good hygiene and hand washing, and wearing a mask if they had respiratory symptoms (*Ministry of Health Singapore, 2020*). Comparatively, by February 9 in Tianjin, 11,700 contacts were under observations and the Baodi district of almost 1 million people was placed under lockdown with restrictions including: one person per household could leave every 2 days to purchase basic needs, public gatherings were banned, no one could leave their homes between 10PM and 6AM without an exemption, entrances to Tianjin were put under control, and all the buses linking nearby provinces and cities were halted (www.chinadaily.com). While Singapore contained the virus spread relatively well until mid-March, they reached 500 confirmed cases on March 23, 1000 cases on April 1, 10,000 cases on April 22, and 25,000 cases on May 13 (*Ministry of Health Singapore, 2020*); Tianjin province began to flatten their epidemic curve by mid-to-late-February and had plateaued at 192 confirmed cases as of May 19 (github.com/CSSEGISandData/COVID-19).

In Singapore and Tianjin we estimated relatively short serial intervals. Of particular note, early estimates of R0 for COVID-19 used the SARS serial interval of 8.4 days (*Abbott et al., 2020*; *Majumder and Mandl, 2020*; *Wu et al., 2020*). Our serial interval findings from two populations mirror those of *Zhao et al., 2020* and (*Nishiura et al., 2020*), who estimated a serial interval of 4.4 and 4.0 days. *Du et al., 2020* obtain a similar estimate for the serial interval (3.96 days with 95% CI: 3.53–4.39) but with standard deviation 4.75 days, based on 468 cases in 18 provinces. Furthermore, we estimate the serial interval to be shorter than the incubation period in both clusters, which suggests pre-symptomatic transmission. This indicates that spread of SARS-CoV-2 is likely to be difficult to stop by isolation of detected cases alone. However, shorter serial intervals also lead to lower estimates of R0, and our serial intervals support R0 values just below 2; if correct this means that half of the transmissions need to be prevented to contain outbreaks.

We stratified the incubation period analysis for Tianjin by time of symptom onset (pre- or post-Jan 31, 2020; motivated by quarantine/social distancing measures) and found that the apparent incubation period was longer for those with post-quarantine symptom onset. The reason for this is unclear, but one possible explanation is that there were (unknown, therefore unreported) exposures during the quarantine period. If people are quarantined in groups of (presumed) uninfected cases, pre-symptomatic transmission in quarantine would result in true exposure times that are more recent than reported last possible exposure times.

Although it may seem contradictory that for example Singapore's efforts were able to keep the epidemic under control using mainly case-based controls if pre-symptomatic transmission is common, it remains the case that detailed contact tracing combined with case finding may be key to

limiting both symptomatic and pre-symptomatic spread. In Singapore, symptom-free close contacts of known cases were quarantined preemptively for 14 days, and other less high-risk contacts were placed under phone surveillance (*Lee et al., 2020*). In addition, if case finding is able to prevent a large portion of symptomatic transmission, it seems logical that the remaining observed transmission may be pre-symptomatic. The large extent of pre-symptomatic spread that is occurring, however, may be one reason that the spread of COVID-19 in Singapore was ultimately only delayed and not prevented.

There are several limitations to this work. First, the times of exposure and the presumed infectors are uncertain, and the incubation period is variable. We have not incorporated uncertainty in the dates of symptom onset. We have used the mixture model approach for serial intervals to avoid assuming that the presumed infector is always the true infector, but the mixture does not capture all possible transmission configurations. Our R0 estimates are simple, based on a doubling time of 4.6 days, and could be refined with more sophisticated modeling in combination with case count data. We have not adjusted for truncation (e.g. shorter serial intervals are likely to be observed first) or the growth curve of the epidemic. However, the serial interval estimates are consistent between the two datasets, are robust to the parameter choices, and are consistently shorter than the estimated incubation times.

We estimated both the incubation period and the serial interval in Singapore and Tianjin COVID-19 clusters. Our results suggest that there is substantial transmission prior to onset of symptoms, as the serial interval is shorter than incubation period by 2–4 days. We find differences in estimated incubation period between early and later cases; this may be due to pre-symptomatic transmission or differences in reporting and/or in perceived exposure as the outbreak progressed, in the context of social distancing measures. Evidence of transmission from apparently healthy individuals makes broad-scale social distancing measures particularly important in controlling the spread of the disease.

## Materials and methods

### Data

All datasets and R code are available on GitHub (github.com/carolinecolijn/ClustersCOVID19; *Tindale et al., 2020*; copy archived at https://github.com/elifesciences-publications/ClustersCOVID19).

Singapore data was obtained from the *Ministry of Health Singapore, 2020* online press releases. The Singapore dataset comprised 93 confirmed cases from the date of the initial case on January 23, 2020 until February 26, 2020. Tianjin data was obtained from the *Tianjin Health Commission, 2020* online press releases. The Tianjin dataset comprises 135 cases confirmed from January 21 to February 22, 2020. The symptom onsets were available on the official website for all but a few patients who had not had symptoms before being diagnosed at a quarantine center. Both datasets contained mainly information on exposure times, contacts among cases, time of symptom onset (See Appendix 1 for column descriptions and data processing).

### Statistical analysis

All statistical analyses were performed using R (*R Development Core Team, 2013*).

#### Incubation periods: not accounting for intermediate cases

The daily incidence of hospitalization and mortality was plotted with the cumulative number of cases confirmed and discharged. The daily incidence was also visualized by date of symptom onset. For the symptom onset plots, any cases that did not have information on date of onset of symptoms were removed. Cases were then grouped based on their assumed source of infection (see Appendix 1 for full details).

Incubation periods were estimated in two ways: directly from the exposure to symptom onset times, and using a model allowing for unknown intermediate cases to have been the true source of infection (see below). The direct estimates were based on the earliest and latest possible exposure times, and on the reported times of symptom onset. It is impossible to confirm the exact times of exposure and thus we used interval censoring, which uses the likelihood of a time falling in a defined

window, (R package icenReg [*Anderson-Bergman, 2017*]) to make parametric estimates of the incubation period distribution. For cases without a known earliest possible exposure time, we assume that the case must have been exposed within the 20 days prior to their symptom onset. For cases without a known latest possible exposure time, we assume that exposure had to have occurred before symptom onset. Some cases had a travel history or contact with a known location or presumed source of the virus and this defined their window for exposure. In the Singapore dataset, other individuals had estimated exposure times based on the symptom times for their presumed infector. For these, we define an exposure window using the symptoms of their presumed infector −7/+4 days. Having defined exposure windows, we proceed with interval censoring. In both datasets we stratified the data according to whether symptom onset occurred early or late, and estimated incubation periods separately. We define 'early' cases as those with symptom onset on or prior to January 31.

## Incubation periods: accounting for intermediate cases

Standard estimates of the incubation period from exposure and symptom data require knowledge of the true exposure event. In our data, exposures were frequently attributed to attendance at events or locations where there had been known COVID-19 cases. It is conceivable that some cases were not in fact exposed at the event, but subsequent to it, by an unknown (perhaps asymptomatic) case who also attended the event or was otherwise linked. We developed the following approach to account for possible unknown intermediates. Suppose the data suggest that case $i$ was exposed at an event by individual $A$, but in fact, there is an unknown intermediate $x$ who was infected at the event and who subsequently infected $i$. In this case, the time between $i$'s apparent exposure and $i$'s symptom onset is not a sample of the incubation period. Instead, it is one generation time (the time between $A$ infecting $x$ and $x$ infecting $i$) followed by one incubation period (from $x$ infecting $i$ to i's symptoms). Similarly, if $x$ infects a second unknown intermediate $y$, and $y$ infects $i$, then the time elapsed is two generation intervals followed by an incubation period. Under the simplifying assumption that the generation time and the incubation period follow a gamma distribution with the same scale parameter, we can explicitly write the density for the elapsed time, given $k$ intermediates. We model the assumption that longer times between (presumed) exposure and symptom onset have more room for undetected intermediate cases. To describe this with likelihoods, we model unknown intermediate cases occurring with a probability proportional to the length of the apparent incubation period, using a Poisson process (see Appendix 1). We estimate the mean incubation period and generation interval with this approach, also accounting for right truncation (which is not available in the interval censoring estimator in icenReg). If $f(t)$, $g(t)$ are the densities for the incubation period and generation time respectively, then with $k$ intermediates, the time elapsed has density $h_k(t) = g * \ldots * g * f = g^{(k)} * f$, where * denotes convolution, i.e., $g * f = \int_0^t g(s)f(t-s)ds$. The right truntction time $T_i$ is the time between $i$'s exposure and the end of the observation period (because if the symptom onset does not happen after $T_i$ has elapsed it will not be observed, and this can bias estimates). Let the time from symptom onset to the beginning of the exposure window be $t_{max}^i$, and to the end of the window $t_{min}^i$. The incubation period is then somewhere in the interval $(t_{min}^i, t_{max}^i)$. The likelihood of observing a time in this interval, conditional on $k$ intermediates, is $L_k^i = \frac{H_k(t_{max}^i) - H_k(t_{min}^i)}{H_k(T_i)}$. We use a Poisson process with rate $r$ to model the probability that there are $k$ intermediates. This means that the likelihood for the $i$'th observation is $L^i = \sum_{k=0}^{3} p_k L_k^i$. The complete likelihood is the product over all cases, $L = \prod_i L^i$. To compute this, note that if $g$ and $f$ are both gamma densities with shapes $a_g$ and $a_i$, and if they have the same scale parameter $b$, then the convolution $g * f(t) = \mathrm{Gamma}(a_g + a_i, b)$. We can extend this to $k$ intermediates: the density is $g^{(k)} * f = \mathrm{Gamma}(k a_g + a_i, b)$. We truncate the number of possible intermediates at $n$, so we condition the usual Poisson probability for $k$ arrivals, $\rho_{k,r} = r^k e^{-r}/k!$, accordingly. Let $C_{n,r} = \sum_{i=0}^{n} p_i(r)$, and use

$$p_k = \begin{cases} \rho_{k,r}/C_{n,r}, & k \leq n \\ 0 & \text{otherwise} \end{cases} \tag{1}$$

We use maximum likelihood to estimate the shape parameters $a_g$ and $a_i$ of the generation and incubation periods under a range of intermediate 'arrival rates' $r$ and we use bootstrapping to

estimate the credible intervals. We refer to this analysis as the 'incubation period with intermediates' analysis.

## Serial intervals: not accounting for intermediates

We illustrate the empirical serial intervals implied by contact links reported in the data. We compute the mean and standard deviation of these in entirety, and separated into early- and late-occurring cases, calculating summary statistics of possible transmission pairs in 'early' (i.e. first date of symptom onset on or before Jan 31, 2020) vs. 'late' portions of both clusters. We estimate the mean serial intervals using these 'directly reported' contacts, accounting for right truncation (R package Surv-Trunc) and using Cox proportional hazards to determine whether there is a significant early vs. late difference. We use the non-parametric survival curves to estimate the mean serial interval for both datasets.

## Serial intervals: accounting for intermediates

As with incubation periods, reported serial intervals may miss unknown intermediates, and co-infectors of two cases presumed to be a transmission pair. We used the expectation-maximization approach described in *Vink et al., 2014*, which not only takes unknown intermediates into account but also explicitly models a fixed set of possible mis-allocation of infector-infectee pairs in contact data. Briefly, this approach assigns the case with earliest symptom onset in a cluster a 'putative index' (PI) status, and uses the time difference between symptom onset of subsequent cases in the cluster and the putative index as 'index case to case' (ICC) intervals for putative index cases in small, closely-linked sets of cases ('small clusters'). The ICC intervals are the time differences between the symptom onset time $t_{pi}$ of the putative index (PI) case and the other members' symptom onset (call these times $t_j$, where $j$ is another case in the same small cluster as this PI). These intervals are not samples of the serial interval distribution, because it need not be the case that the PI infected the others. *Vink et al., 2014* used a mixture model in which ICC intervals $t_j - t_{pi}$ can arise in four ways: (1) an outside case infects PI and $j$; (2) PI infects $j$; (3) PI infects an unknown who infects $j$ and (4) PI infects unknown one who infects unknown two who infects $j$. Accordingly, if the serial interval $x \sim \mathcal{N}(\mu, \sigma^2)$, the density for the ICC intervals is

$$f(x; \mu, \sigma^2) = \sum_i w_i f_i(\mu, \sigma^2)$$

where $w_i$ are weights of the $i$'th component density and $f_i$ are the component densities for the $i$'th transmission route. Expectation-maximization is used to determine $\mu$ and $\sigma$ (See *Vink et al., 2014* for more details).

For each dataset, we create a network, with individuals represented by nodes. The network's edges are the reported direct contacts between individuals. Every such network (or graph) consists of one or more components – sets of nodes that are connected by edges. We use the components of the network to define transmission clusters. Since the four models in the mixture are likely insufficient to model the transmission in large clusters, we restrict the analysis to only the first four cases per cluster (or the first 3, 5, or 6 cases per cluster to determine impact of altering number of cases per cluster; see Appendix 1). We defined the first case within the cluster as the case with the earliest date of symptom onset within the cluster; however we also examined the impact of using the earliest end exposure time if the first symptomatic case was not the index case for the cluster (See Appendix 1). Given the serial interval, we calculate an approximate reproduction number using the empirical growth rate (*Wallinga and Lipsitch, 2007*): $R = \exp{(r\mu - 1/2 r^2 \sigma^2)}$, where $r$, $\mu$ and $\sigma$ are the exponential growth rate, the mean serial interval and the standard deviation of the serial interval, respectively). To obtain confidence intervals for $R$ we resample $\mu$ and $\sigma$ using bootstrapping.

## Pre-symptomatic transmission

We estimate the portion of transmission that occurs before symptoms as the fraction of samples where serial interval minus incubation period is negative. We introduce an approach to take covariation between the two variables into account, as follows. The mean difference between two random variables is the difference between the means. Therefore, the mean serial interval minus the mean incubation periods gives an estimate of the mean time before symptoms that transmission occurs

according to our data. However, the distribution of the difference depends on the covariance between the incubation period and the serial interval. Unfortunately, it is challenging to obtain a good estimate of the covariance between these quantities. We estimated the covariance (and correlation) using case pairs; each pair is associated with two numbers: a serial interval estimate and an incubation period for the infectee. The covariance between these is a (somewhat crude) estimate of the covariance in the incubation period and serial intervals. We sampled incubation periods and serial intervals from our estimated distributions, ensuring that we respected the observed correlation, and used the serial interval - incubation period differences to estimate the portion of transmission that is pre-symptomatic. Further details of this approach are given in Appendix 1.

In estimating pre-symptomatic transmission, we compare 'direct' (not accounting for intermediates) incubation periods and serial intervals, and we compare the two *with* accounting for intermediates. We take the covariation into account throughout.

## Acknowledgements

We thank the Ministry of Health Singapore and the Tianjin Health Commission for publishing information about cases through online press releases. We thank the participants of 'EpiCoronaHack' at Simon Fraser University for their roles in curating these two datasets. We thank the public health teams and the patients whose data have been included in these analyses.

## Additional information

### Competing interests

Caroline Colijn: Reviewing editor, *eLife*. Jacco Wallinga: Reviewing editor, *eLife*. The other authors declare that no competing interests exist.

### Funding

| Funder | Grant reference number | Author |
| --- | --- | --- |
| Government of Canada | Canada 150 Research Chair program | Caroline Colijn |

The funders had no role in study design, data collection and interpretation, or the decision to submit the work for publication.

### Author contributions

Lauren C Tindale, Data curation, Validation, Visualization, Methodology, Writing - original draft, Writing - review and editing; Jessica E Stockdale, Michelle Coombe, Formal analysis, Visualization, Methodology, Writing - review and editing; Emma S Garlock, Visualization, Writing - review and editing; Wing Yin Venus Lau, Data curation, Visualization, Writing - review and editing; Manu Saraswat, Formal analysis; Louxin Zhang, Jacco Wallinga, Data curation, Writing - review and editing; Dongxuan Chen, Data curation; Caroline Colijn, Conceptualization, Software, Formal analysis, Supervision, Validation, Investigation, Methodology, Writing - original draft, Project administration, Writing - review and editing

### Author ORCIDs

Lauren C Tindale https://orcid.org/0000-0001-7751-1042
Jessica E Stockdale https://orcid.org/0000-0001-7984-1010
Louxin Zhang http://orcid.org/0000-0003-0260-824X
Caroline Colijn https://orcid.org/0000-0001-6097-6708

### Decision letter and Author response

Decision letter https://doi.org/10.7554/eLife.57149.sa1
Author response https://doi.org/10.7554/eLife.57149.sa2

## Additional files

### Supplementary files

• Transparent reporting form

### Data availability

Data are available on github at https://github.com/carolinecolijn/ClustersCOVID19 (copy archived at https://github.com/elifesciences-publications/ClustersCOVID19). Code to produce all analyses is also available there. Source data files of the Singapore and Tianjin clusters have been provided.

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

## Appendix 1

# Details of the Singapore and Tianjin datasets

### The Singapore dataset

In the Singapore dataset: 'related cases' are direct known contacts between cases; 'cluster links' are cases that are linked together through an identified cluster event; 'presumed infected date' and 'presumed reason' are the earliest known date and the reason that each case was known to likely be infected; 'last poss exposure' and 'contact based exposure' are sub-classifications of 'presumed infected date', representing either the last date that each case could have been infected – the date of arrival in Singapore for travellers from Wuhan – or the date that each case was likely infected during a local transmission event in Singapore, respectively; 'cluster' is the Ministry of Health Singapore's classification of cases into transmission cluster events.

These data inform the ''start_source'' and ''end_source'' columns which encode the earliest and latest possible dates of case exposure. For example, we assume that those travelling from Wuhan were exposed before travel (due to evidence of lack of community transmission in Singapore at the time), and that those cases associated with a particular event or location (e. g., Grace Assembly gatherings, the visit to the Yong Thai store by a tour group from Wuhan) were not exposed prior to that event. For the latter, we set ''end_source'' to the date of the event +4 days, to allow for some uncertainty and the possibility of an intermediate infector. For cluster cases thought to originate from a particular index case but lacking information on dates of contact, ''start_source'' is set to the first symptom onset in the cluster -7 days. The ''end_source'' is set assuming that once a case in a cluster was identified, people were well aware of this and ceased mixing within the group; thus ''end_source'' is the minimum of the earliest quarantine, hospitalization or symptom onset in the cluster, and the symptom onset date of the case in question.

In the absence of other information, we set the ''start_source'' of a case to their symptom onset date - 20 days (to allow for a wide range of epidemiologically feasible incubation periods), and the ''end_source'' to their symptom onset (since all cases must be exposed before they show symptoms).

All cases in the Singapore dataset were categorized into an infection source group based on information provided in the 'presumed reason' column without conflict. The group designations were not used in the statistical estimates.

### The Tianjin dataset

In the Tianjin cases summary spreadsheet, the main columns are: gender, age, symptom onset, symptom type, confirmation date, severity and death date *Tianjin Health Commission, 2020*; detailed information from daily reports for the first 80 patients provided travel or exposure history and contact information, from which we obtained exposure windows (start source, end source). For backup and to complete missing information for later cases we also referred to *Jinyun News, 2020*, Tianjin official local media, who used Baodi local government reports (*Tianjin Baodi People's Government, 2020*). They reported detailed activity for those confirmed cases when their corresponding epidemiological history investigation was finished.

The ''start_source'' and ''end_source'' columns were defined similarly to the Singapore dataset where possible, with reasoning provided in the ''Infection_source'' column and further explanation in ''recorrection for start and end source''. In most cases, start and end times in the Tianjin dataset were defined by known windows of contact with other individuals with confirmed COVID-19 infections, the Baodi shopping mall or travel to areas with higher levels of infection such as Wuhan. Again, in the absence of other information, we set the ''start_source'' of a case to their symptom onset date -20 days and the ''end_source'' to their symptom onset.

Cases in the Tianjin dataset were categorized into an infection source group based on information provided in the the ''Infection source'' column. There were a small number of

cases (n = 12) that could be classified into two possible infection source groups (e.g. from Wuhan and has a close relationship with another known case). These cases were assigned their infection source groups based on the following hierarchy of possible sources: (highest priority) Wuhan or Hubei origin > Mall (for shoppers, workers, or individuals living near to the Baodi mall outbreak) > Family relationship > Work relationship > Other known relationship > Other travel > Unknown (lowest priority).

## Statistical methods

### Incubation period

The ''start_source'' and ''end_source'' columns in each dataset are used to define the maximum and minimum possible incubation periods for each case. We additionally assume that incubation times have to be at least 1 day in length, and that the maximum incubation times are at least 3 days to take into account some uncertainty on symptom onset reporting.

We explored several distributions for the incubation period: gamma, Weibull and log normal. As shown in *Figure 4*, once fit the resulting distributions all provide very similar results. *Appendix 1—table 1* summarizes the parameter estimates for these three distributions. *Appendix 1—table 2* gives the parameters for the incubation period for early- and late-occurring cases in both datasets.

**Appendix 1—table 1.** Incubation period estimates (without intermediates) using gamma, Weibull and log normal distributions. 95% confidence intervals for the shape and scale (log mean and sd for log normal) parameters are shown in brackets.

| Gamma | Median | Shape | Scale |
|---|---|---|---|
| Singapore Cluster | 5.32 | 3.05 (2.0, 3.84) | 1.95 (1.23, 2.34) |
| Tianjin Cluster | 8.06 | 4.74 (3.35, 5.72) | 1.83 (1.29, 2.04) |

| Weibull | Median | Shape | Scale |
|---|---|---|---|
| Singapore Cluster | 5.66 | 1.83 (1.45, 2.30) | 6.91 (5.77, 8.29) |
| Tianjin Cluster | 8.59 | 2.41 (1.99, 2.90) | 10.01 (8.94, 11.20) |

| Log normal | Median | Log mean | Standard deviation |
|---|---|---|---|
| Singapore Cluster | 4.83 | 1.57 (1.38, 1.81) (mean 4.81) | 0.60 (0.47, 0.76) |
| Tianjin Cluster | 7.66 | 2.04 (1.92, 2.22) (mean 7.69) | 0.47 (0.39, 0.56) |

**Appendix 1—table 2.** Incubation period estimates (without intermediates) using stratified data

**Tianjin**

| Gamma | Median | Shape | Scale |
|---|---|---|---|
| Early | 6.48 | 6.01 (3.61, 7.26) | 1.140 (0.66,1.276) |
| Late | 12.1 | 17.78 (9.52, 21.47) | 0.695 (0.379,0.778) |
| **Weibull** | **Median** | **Shape** | **Scale** |
| Early | 6.73 | 2.88 (2.16, 3.48) | 7.643 (6.735, 8.553) |
| Late | 12.6 | 4.34 (3.10, 5.24) | 13.661 (12.245, 15.289) |
| **Log normal** | **Median** | **Log mean** | **Standard deviation** |
| Early | 6.30 | 1.84 (1.70,2.03) | 0.426 (0.331,0.547) |
| Late | 12.0 | 2.48 (2.38,2.67) | 0.233 (0.172,0.315) |

**Singapore**

| Gamma | Median | Shape | Scale |
|---|---|---|---|
| Early | 5.26 | 3.22 (1.67, 4.05) | 1.818 (0.847,2.18) |
| Late | 5.35 | 2.96 (1.68,3.72) | 2.034 (1.132,2.439) |
| **Weibull** | **Median** | **Shape** | **Scale** |
| Early | 5.51 | 2.05 (1.34,2.58) | 6.587 (5.077,7.897) |
| Late | 5.67 | 1.75 (1.29,2.21) | 6.989 (5.408,8.38) |
| **Log normal** | **Median** | **Log mean** | **Standard deviation** |
| Early | 4.91 | 1.59 (1.33,1.82) | 0.598 (0.421,0.848) |
| Late | 4.72 | 1.55 (1.25,1.78) | 0.606 (0.441,0.834) |

## Serial interval

We used bootstrapping to explore the range for the point estimates of $\mu$ and $\sigma$ from the mixture model. *Appendix 1—figure 1* shows the results. The mean of the bootstrapped mean estimates is $4.49 \pm 0.716$ for Tianjin and $3.83 \pm 0.882$ days in Singapore. Bootstrap values are consistent with a serial interval that is considerably shorter than the incubation periods in both datasets. *Appendix 1—table 3* shows the sensitivity analysis; we varied the the number of cases per cluster to include in the ICC interval data and we explored sorting the cases in the clusters according to the time of last exposure (i.e., the putative index status assigned to the individual with the earliest end to their exposure window, instead of the first symptomatic individual).

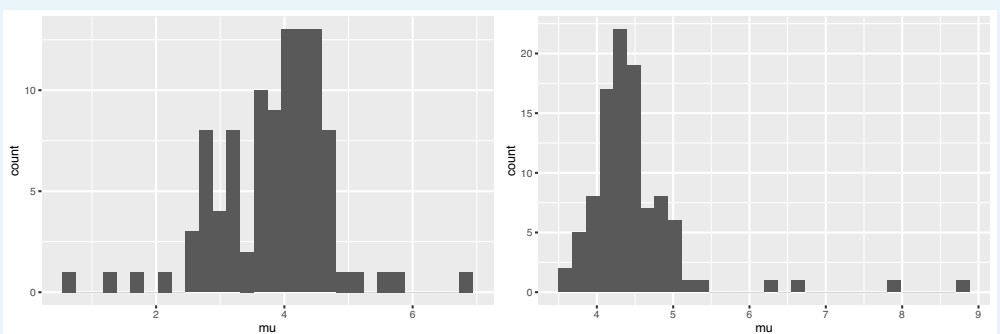

**Appendix 1—figure 1.** Bootstrap values of the mean serial interval for (left) Singapore and (right) Tianjin, based on 100 replicates using the first four cases in each cluster.

**Appendix 1—table 3.** Serial interval estimates: accounting for intermediates.

| ordering | Number cases per cluster | $\mu$ (Tianjin) | $\sigma$ (Tianjin) | $\mu$ (Singapore) | $\sigma$ (Singapore) |
|---|---|---|---|---|---|
| Onset | 3 | 4.17 | 0.998 | 4.03 | 1.06 |
| Onset | 4 | 4.31 | 0.935 | 4.17 | 1.06 |
| Onset | 5 | 4.43 | 0.999 | 4.43 | 1.09 |
| Onset | 6 | 4.54 | 1.05 | 4.76 | 1.15 |
| Last Exposure | 4 | 5.09 | 1.27 | 4.26 | 1.17 |
| Bootstrap | 4 | 4.49 (sd 0.716) | 0.995 (sd 0.307) | 3.83 (sd 0.882) | 1.24 (sd 0.538) |

The primary analysis removes all cases that are missing dates of symptom onset. To explore the potential impact of removing cases we repeated the serial interval estimates—when accounting for intermediates—by including these missing cases with imputed dates of symptom onset. There are 10 cases with missing date of symptom onset in both Tianjin and Singapore datasets. All cases missing date of symptom onset have a date of confirmation for infection with SARS-CoV-2; therefore, imputed dates were calculated by: (date of confirmation for case) - (average difference between date of symptom onset and date of confirmation, for all cases used in main analysis). This average difference between date of symptom onset and date of confirmation is 5.23 days in Tianjin and 7.43 days in Singapore. Imputing dates in this manner assumes that dates of symptom onset are missing completely at random. This assumption seems reasonable as the range of date of confirmation for the 10 imputed cases covers the majority of the range of date of confirmation for cases in the main analysis, in both datasets (Feb one to Feb 22, 2020 for imputed cases vs. Jan 21 to Feb 22, 2020 for main analysis cases in Tianjin and Jan 31 to Feb 21, 2020 vs. Jan 23 to Feb 26, 2020 in Singapore). *Appendix 1—table 4* contains the results of serial interval estimates including cases with imputed date of symptom onset and demonstrates that there is no substantial difference compared to serial interval estimates from the main analysis where missing cases are removed (*Appendix 1—table 3*).

**Appendix 1—table 4.** Serial interval estimates: accounting for intermediates and using imputed dates of symptom onset

| ordering | Number cases per cluster | $\mu$ (Tianjin) | $\sigma$ (Tianjin) | $\mu$ (Singapore) | $\sigma$ (Singapore) |
|---|---|---|---|---|---|
| Onset | 3 | 4.35 | 0.907 | 4.18 | 1.05 |
| Onset | 4 | 4.40 | 0.864 | 4.27 | 1.04 |
| Onset | 5 | 4.48 | 0.909 | 4.41 | 0.981 |
| Onset | 6 | 4.55 | 0.948 | 4.71 | 1.08 |
| Last Exposure | 4 | 4.81 | 0.948 | 4.62 | 2.11 |
| Bootstrap | 4 | 4.53 (sd 0.585) | 0.941 (sd 0.358) | 4.31 (sd 1.03) | 1.50 (sd 0.629) |

## Pre-symptomatic transmission: methods details

We estimated the portion pre-symptomatic transmission taking into account that the serial interval and incubation period are not independent, as described in the main text. In Singapore, we found that the covariance is 5.88, the Pearson correlation is 0.43 ($p = 0.001$) and the Spearman (rho = 0.174) and Kendall (tau = 0.134) correlations were not significant ($p = 0.2$); this is an intermediate signal of covariation. In Tianjin the covariance was 2.63, the correlation 0.29 and the statistical signal similar. We used both our 'direct' and

'intermediate' incubation period analysis to determine the portion pre-symptomatic transmission, accounting for the covariation.

To do this, we first sampled the incubation period parameters using the fits to data in the main text. These fits include a variance estimate between the shape and scale parameters, so we sample the shape and scale accordingly (using the gamma distribution). We created 100 incubation period (shape, scale) pairs (i.e., 100 samples). There are samplers in R for multivariate distributions whose margins are both gamma (rmvgamma in the lcmix package) and of course multivariate normal samplers, but we do not have a sampler for jointly distributed random variables with a normal distribution on one margin and a gamma on the other. Therefore, we use a gamma distribution for the serial interval, with the same mean and variance as the normal distribution estimated directly from the case-pair data. We obtain 100 serial interval gamma (shape, scale) pairs with the appropriate mean and variance. For each of these 100 distributions, sample *jointly* 500 incubation periods and serial intervals, with correlation of approximately 0.3. We therefore have $100 \times 500 = 50,000$ joint samples of incubation period and serial interval. The fraction of the (serial interval minus incubation period) samples is an estimate of the fraction of transmission that is pre-symptomatic, accounting for covariation.

We take the same approach for the estimates that account for intermediates; therefore, we sample from the gamma distribution for the incubation period as estimated with intermediates, from the ICC estimate of the serial intervals (i.e., we sample 100 incubation period shape, scale pairs, and 100 generation time shape, scale pairs, and for each we create 500 samples of the incubation period and generation time, accounting for covariance). This yields the estimates in *Table 1* and the density plots in *Figure 8*.

## Additional published estimates

Estimates of incubation period and serial interval from other studies are shown in *Appendix 1—table 5*. Of note, the majority of studies do not estimate both incubation period and serial interval in the same population.

**Appendix 1—table 5.** Mean incubation period and mean serial interval estimates for COVID-19 generated by other studies.

| Data | Number of Cases | Mean Incubation Period (days) | Mean Serial Interval (days) | Reference |
|---|---|---|---|---|
| Wuhan first cases | 425 | 5.2 (95CI 4.1-7.0) | 7.5 (95CI 5.3-19) | *Li et al., 2020b* |
| South Korea first cases | 24 | 3.6 | 4.6 | *Ki and Task Force for 2019-nCoV, 2020* |
| Travellers from Wuhan | 88 | 6.4 (95CI 5.6-7.7) | - | *Backer et al., 2019* |
| Diagnosis outside Wuhan (excluding Wuhan residents) | 52 | 5.0 (95CI 4.2-6.0) | - | *Linton et al., 2020* |
| Diagnosis outside Wuhan (including Wuhan residents) | 158 | 5.6 (95CI 5.0-6.3) | - | *Linton et al., 2020* |
| Transmission chains in Hong Kong | 21 chains | - | 4.4 (95CI 2.9-6.7) | *Zhao et al., 2020* |
| Infector-infectee pairs* | 28 pairs | - | 4.0 (95CI 3.1-4.9) | *Nishiura et al., 2020* |

*Note: included 3 infector-infectee pairs from this Singapore cluster. Remainder from Vietnam (4), South Korea (7), Germany (4), Taiwan (1) and China (9).

