## [Decision Letter]

**Acceptance summary:**

This is a valuable contribution to the effort to understand the timing of transmission of COVID-19 relative to symptoms. It provides evidence that pre-symptomatic individuals contribute to transmission and thus efforts must be directed beyond simply isolating those who have symptoms. This study provides data in support of widespread adoption of social distancing and other transmission prevention measures.

**Decision letter after peer review:**

Thank you for submitting your article "Evidence for spread of COVID-19 prior to symptom onset supports the need for social distancing" for consideration by *eLife*. Your article has been reviewed by three peer reviewers, including Marc Lipsitch as the Reviewing Editor and Reviewer #1, and the evaluation has been overseen by Eduardo Franco as the Senior Editor. The following individuals involved in review of your submission have agreed to reveal their identity: Joel Miller (Reviewer #2); Virginia E. Pitzer (Reviewer #3).

The reviewers have discussed the reviews with one another and the Reviewing Editor has drafted this decision to help you prepare a revised submission.

As the editors have judged that your manuscript is of interest, but as described below that additional explanation and possibly new analyses are required before it is published, we would like to draw your attention to changes in our revision policy that we have made in response to COVID-19 (https://elifesciences.org/articles/57162). First, because many researchers have temporarily lost access to the labs, we will give authors as much time as they need to submit revised manuscripts. We are also offering, if you choose, to post the manuscript to bioRxiv (if it is not already there) along with this decision letter and a formal designation that the manuscript is 'in revision at *eLife*'. Please let us know if you would like to pursue this option. (If your work is more suitable for medRxiv, you will need to post the preprint yourself, as the mechanisms for us to do so are still in development.)

Summary:

The manuscript by Tindale et al. presents estimates of the incubation period and serial interval for SARS-CoV-2 based on publicly available datasets from Singapore and Tianjin, China, where control measures were applied early and there was well documented contract tracing. Based on their findings that estimates of the mean incubation period are longer than estimates of the serial interval, they conclude that substantial pre-symptomatic transmission must be occurring.

Essential revisions:

1) The SI should shorten as control measures become more stringent (finding contacts faster), but this is not discussed, only the slightly odd lengthening of the incubation period. For example such shortening was prominent in Lipsitch et al., 2003 for SARS. Comment please.

2) Ferguson et al. (Imperial College COVID Report 4) have noted a previously unappreciated bias in retrospective contact tracing that can inflate the number of short serial intervals or incubation period. The relevance is uncertain to me for this study, but should be addressed.

3) The data do indeed seem to support presymptomatic transmission to a significant degree. Can the authors be any more specific about the bounds on this? For example if the serial intervals and incubation periods were uncorrelated, or positively correlated, or if one estimated for each case a probability that transmission was before symptoms, what would the result be in terms of the actual proportion of transmission presymptomatic?

4) Assuming the answer to 3 is "a lot" as I suspect, there is a paradox here – that Singapore kept the epidemic under control for months with mainly case-based interventions before it failed, a situation that could not happen (I think) if most transmission was presymptomatic. These two facts (the analysis and the control that lasted until early April) seem to contradict each other. Please discuss.

5) We would like a little more discussion on the lengthening of the incubation period – could this simply mean that before the restrictions the interaction that is identified as transmitting wasn't responsible, but in fact the transmission had happened days earlier, perhaps in an earlier interaction of the two individuals? It would be nice if the authors can comment on why we can or cannot rule this possibility out.

6) A key concern is that, unlike the method used to estimate the serial interval, the approach used to estimate the incubation period assumes that all cases (and potential infectors) have been identified and confirmed, and does not allow for transmission from asymptomatic cases. This seems particularly problematic for the Singapore dataset, in which all of the data seems to be on hospitalized cases; thus, less severe cases may have been missed (unless all cases were hospitalized as a form of isolation there). In Tianjin, it is unclear whether confirmed cases were hospitalized or not, but seems likely that asymptomatic or mild infections may have been missed. The large increase in the estimated incubation period for the "early" vs. "late" cases suggests to me that exposure to asymptomatic/undetected cases is likely biasing the results, as one would expect a greater prevalence and thus higher risk of exposure to asymptomatic cases later on. If there was an undetected case in between a confirmed case and the purported index case, presumably the estimated incubation period would be reduced by ~50% (at least).

7) The method for estimating the serial interval, on the other hand, does allow for multiple (undetected) generations of transmission between confirmed cases in the mixture model. Thus, these two approaches make different assumption about the potential presence/role of asymptomatic cases. Furthermore, there is no real methods development here. The authors apply two different established methods to their data, rather than presenting a unified framework for estimating the incubation period and serial interval (which would be novel and considerably strengthen the analysis).

8) Lastly, the estimates of R0 that they present in the Abstract and main text are not based on the actual data from Singapore and Tianjin. Instead, they use their estimates of the serial interval mean and standard deviation for each setting and a point estimate of the growth rate (with no uncertainty) from a different study based on data for all of mainland China. Thus, it is misleading to suggest that these are the R0 estimates specifically for Singapore and Tianjin.

---

## [Author Response]

Essential revisions:1) The SI should shorten as control measures become more stringent (finding contacts faster), but this is not discussed, only the slightly odd lengthening of the incubation period, For example such shortening was prominent in Lipsitch et al., 2003 for SARS. Comment please.

We have added Cleveland-style figures and compared serial intervals between the early and late phases (Figure 6), as well as adding a discussion of the reviewer’s point to the text (“Serial Intervals” section of the Results). The raw serial intervals do seem to shorten by about a day in both locations in later vs. early subsets of the data. We have also added survival analysis exploring whether this result might be due to right truncation. This did not find a significant difference between the early and late stages, though this does not rule out that shortening occurred. We have cited the Science paper.

2) Ferguson et al. (Imperial College COVID Report 4) have noted a previously unappreciated bias in retrospective contact tracing that can inflate the number of short serial intervals or incubation period. The relevance is uncertain to me for this study, but should be addressed.

Report 4 is an analysis of the case fatality rate and so we are not certain what the previously unappreciated bias is. We agree that contact tracing can be biased in a number of ways, including towards close contacts who individuals are likely to be able to name, recollected exposures more likely to be recent ones, and right truncation – the fact that individuals with earlier symptom onset are observed sooner (though this will affect both serial intervals and incubation periods). We acknowledge in Appendix 1 that we have not accounted for these effects, and have commented on this in the Discussion. We also adjusted for right truncation in our new analysis accounting for intermediate cases in the incubation period estimates (see below).

3) The data do indeed seem to support presymptomatic transmission to a significant degree. Can the authors be any more specific about the bounds on this? For example if the serial intervals and incubation periods were uncorrelated, or positively correlated, or if one estimated for each case a probability that transmission was before symptoms, what would the result be in terms of the actual proportion of transmission presymptomatic?

This is a challenging and interesting question. In fact one of the reasons that we didn’t highlight our pre-symptomatic fractions more in the first version was that we were aware that subtracting independent samples from two distributions does not give a good estimate of the distribution of differences (only the difference in mean, by linearity). We have developed this aspect of the work considerably, and we now account for correlations. To do this, we estimated the covariance (and correlation) of the serial interval and incubation period distribution using the “direct” samples, i.e. as taken as directly as possible from the data. We then resampled jointly from our estimated distributions, accounting for this level of covariation. Surprisingly, this did not make as much difference to the portion of samples of (serial interval minus incubation period) that are negative as we would have guessed. These remain high, and are higher in Tianjin than in Singapore. The analysis is at the end of the Materials and methods and Results section and the sampling method (based on multivariate distributions with marginal gamma densities) is given in Appendix 1, subsection “Pre-symptomatic transmission: methods details”.

4) Assuming the answer to 3 is "a lot" as I suspect, there is a paradox here – that Singapore kept the epidemic under control for months with mainly case-based interventions before it failed, a situation that could not happen (I think) if most transmission was presymptomatic. These two facts (the analysis and the control that lasted until early April) seem to contradict each other. Please discuss.

The answer to 3 does appear to be “quite a lot” – about half or even more in the new Table 1. We agree that this is a good point and have added details to the Discussion. Essentially, we do not think that pre-symptomatic transmission necessarily renders case-based interventions ineffective, so long as contact tracing is thoroughly implemented and is complemented with strong case finding. In Singapore, symptom-free close contacts were preemptively quarantined – we suspect this minimized community presymptomatic transmission once a cluster is identified. We cite Lee, Chiew and Khong, 2020, for containment efforts.

Also, if effective case finding limits the amount of symptomatic transmission that occurs, it makes sense that the proportion of presymptomatic spread we find would be higher, since this represents the remaining transmission that was not prevented.

5) We would like a little more discussion on the lengthening of the incubation period – could this simply mean that before the restrictions the interaction that is identified as transmitting wasn't responsible, but in fact the transmission had happened days earlier, perhaps in an earlier interaction of the two individuals? It would be nice if the authors can comment on why we can or cannot rule this possibility out.

Yes, we suspect that this must be to do with the interactions assumed to be responsible for exposure being mis-allocated for Tianjin. We have updated the data and some of the assumptions around exposure and we now find that the early-late difference is only significant in Tianjin. In fact this is consistent with pre- or asymptomatic transmission; if individuals were exposed to someone with symptoms, and then later exposed to an individual without symptoms (for example in a group quarantine setting), the exposure would be likely attributed to pre-quarantine contact and the time interval in the data would not be a sample of the incubation period. Instead, it would be a sample of one or more generation times plus an incubation period – see below. We have added discussion of this in the incubation period Results section.

6) A key concern is that, unlike the method used to estimate the serial interval, the approach used to estimate the incubation period assumes that all cases (and potential infectors) have been identified and confirmed, and does not allow for transmission from asymptomatic cases. This seems particularly problematic for the Singapore dataset, in which all of the data seems to be on hospitalized cases; thus, less severe cases may have been missed (unless all cases were hospitalized as a form of isolation there). In Tianjin, it is unclear whether confirmed cases were hospitalized or not, but seems likely that asymptomatic or mild infections may have been missed. The large increase in the estimated incubation period for the "early" vs. "late" cases suggests to me that exposure to asymptomatic/undetected cases is likely biasing the results, as one would expect a greater prevalence and thus higher risk of exposure to asymptomatic cases later on. If there was an undetected case in between a confirmed case and the purported index case, presumably the estimated incubation period would be reduced by ~50% (at least).

This is a good point. Please see the response to the related point :

7) The method for estimating the serial interval, on the other hand, does allow for multiple (undetected) generations of transmission between confirmed cases in the mixture model. Thus, these two approaches make different assumption about the potential presence/role of asymptomatic cases. Furthermore, there is no real methods development here. The authors apply two different established methods to their data, rather than presenting a unified framework for estimating the incubation period and serial interval (which would be novel and considerably strengthen the analysis).

We have introduced another analysis with what we believe is a novel method. As noted above, if there were unknown intermediates, then the sampled “incubation period” would instead be a sample of a sum of generation intervals: A infects B, B infects C, … followed by an incubation period: C’s time from exposure to symptom onset. Under the assumption that the incubation period and generation time can be modelled as a gamma distribution, it is possible to write down the likelihood for the time from A’s exposure (at an event, say, or mass gathering) and C’s symptom onset, because the density is a convolution of gamma densities. We have developed a new analysis along these lines, and we do indeed find shorter incubation periods than without intermediates. We also find that the incubation period is longer than the generation time, giving an alternative point of support for the conclusion about pre-symptomatic transmission. To our knowledge this, combined with the covariance analysis described above, is a novel methodological contribution.

8) Lastly, the estimates of R0 that they present in the Abstract and main text are not based on the actual data from Singapore and Tianjin. Instead, they use their estimates of the serial interval mean and standard deviation for each setting and a point estimate of the growth rate (with no uncertainty) from a different study based on data for all of mainland China. Thus, it is misleading to suggest that these are the R0 estimates specifically for Singapore and Tianjin.

We agree and have modified the wording. R0 estimation (in any sense, never mind for specific populations) was never a primary focus of the work and this section is only there to note that models using SARS-like serial intervals (7.5 days) would conclude that R0 was higher than models using our serial interval estimates of 4+ days. We have amended this.